



**Quantifying the effect of aerosol on vertical velocity and effective terminal**
**velocity in warm convective clouds**
**Guy Dagan, Ilan Koren\*, and Orit Altaratz**
Department of Earth and Planetary Sciences, The Weizmann Institute of Science,
Rehovot 76100, Israel
\*Corresponding author. E-mail: ilan.koren@weizmann.ac.il
**Abstract**
Better representation of cloud–aerosol interactions is crucial for an improved
understanding of natural and anthropogenic effects on climate. Recent studies have
shown that the overall effect can be viewed as a competition between processes with
opposing trends. Here, we reduce the system's dimensions to its center of gravity
(COG), enabling distillation and simplification of the overall trend and its temporal
evolution. Within the COG framework, we show that the aerosol effects are nicely
reflected by the interplay of the system's characteristic vertical velocities, namely the
updraft ($w$) and the effective terminal velocity ($\eta$). The system's vertical velocities can
be regarded as a sensitive measure for the evolution of the overall trends with time.
Using bin-microphysics cloud-scale model, we analyze and follow the trends of the
aerosol effect on the magnitude and timing of $w$ and $\eta$, and therefore the overall
vertical COG velocity. Large eddy simulation model runs are used to upscale the
analyzed trends to the cloud-field scale and study how the aerosol effects on temporal
evolution of the field's thermodynamic properties are reflected by the interplay
between the two velocities. Our results suggest that aerosol effects on air vertical
motion and droplet mobility imply an effect on the way in which water is distributed
along the atmospheric column. Moreover, the interplay between $w$ and $\eta$ predicts the
overall trend of the field's thermodynamic instability. These factors have an important
effect on the local energy balance.



## 1. Introduction


Clouds are key players in the Earth's climate system via their influence on the energy
balance (Baker and Peter, 2008;Trenberth et al., 2009) and hydrological cycle. Of all
of the anthropogenic effects on climate, aerosol's effect on clouds remains one of the
most uncertain (Boucher et al., 2013). In warm clouds, aerosol act as cloud
condensation nuclei (CCN) around which droplets can form, and therefore aerosol
amount and properties determine the initial number of droplets and their size
distribution (Squires, 1958;Rosenfeld and Lensky, 1998;Andreae et al., 2004;Koren et
al., 2005). The initial droplet concentration affects cloud dynamics via microphysical
and dynamical feedback throughout their lifetime. For example, the onset of
significant collision events between droplets in polluted clouds (which are initially
smaller and more numerous than in clean clouds (Squires, 1958)) is delayed (Gunn
and Phillips, 1957;Rosenfeld, 1999, 2000;Squires, 1958;Warner, 1968). This delay
can have opposing effects on cloud development by increasing both the water loading
(which reduces cloud buoyancy and vertical development) and the latent heat release
resulting from the longer and more efficient condensation (increasing cloud buoyancy
and vertical development) (Dagan et al., 2015a;Dagan et al., 2015b;Pinsky et al.,
2013;Koren et al., 2014). We note that often, these opposing effects act at different
stages of the cloud's lifetime, further complicating the prediction of overall trends.
Vertical velocities ($w$) are among the key processes driving convective clouds. The
intensity, duration and characteristic size of the updrafts determine the convective
clouds' properties. In addition, the clouds' vertical velocity affects the distribution of
water along the atmospheric column, thereby having a strong effect on radiation
(Koren et al., 2010) and heat balance (Khain et al., 2005). Although previous studies
have focused on deep convective clouds, these effects are expected to be significant in
warm convective clouds as well. Moreover, warm processes serve as the initial and
boundary conditions for mixed-phase processes in deep convective clouds, and
therefore gaining a better process understanding of the warm phase is essential for
understanding the deeper systems (Chen et al., 2017).
The system has another characteristic velocity that measures droplet mobility. This
velocity, defined as the effective terminal velocity ($\eta$), measures the weighted-by-
mass terminal velocity of all hydrometeors within a given volume and therefore



defines the falling velocity of the volume's center of gravity (COG) (Koren et al.,
2009;Koren et al., 2015) compared to the air vertical velocity. Smaller droplets imply
smaller $|\eta|$ (higher mobility) and therefore less deviation from the surrounding air
movement. Since $\eta$ is always negative, smaller $|\eta|$ implies that per a given air updraft,
the collective liquid water mass will be carried up higher in the atmosphere. The
movement of the COG compared to the surface, defined as $V_{COG,}$ is the vector sum of
the two velocities: $V_{COG} = w - |\eta|$.
$V_{COG}$ has recently been shown to be a good measure for the temporal evolution of
thermodynamic instability in cloud fields (Dagan et al., 2016). $V_{COG}$ represents the
vertical movement of liquid water, which is downward gradient of the net
condensation-less-evaporation profile. A negative $V_{COG}$ implies net transport of the
liquid water from the cloudy layer to the sub-cloud layer. This holds true for clean
precipitation conditions (Dagan et al., 2016), in which the water that condenses in the
cloudy layer sediments down to the sub-cloud layer where it partially evaporates. The
net condensation in the cloudy layer and the net evaporation in the sub-cloud layer
produce a decrease in the thermodynamic instability with time. On the other hand, for
the polluted non-precipitating cases, $V_{COG}$ is positive, indicating that the net liquid
water movement is upward. The water that is being condensed in the lower part of the
cloudy layer is transported upward and evaporates in the upper cloudy and inversion
layers (Dagan et al., 2016). The end result of this vertical condensation–evaporation
profile is an increase in thermodynamic instability with time.
Khain et al. (2005) used a two-dimensional cloud model with spectral (bin)
microphysics to study the aerosol effect on deep convective cloud dynamics. They
concluded that one of the reasons for comparatively low $w$ in clean maritime
convective clouds compared to polluted continental ones is the rapid creation of
raindrops. This increases the liquid water loading in the lower part of the cloud,
thereby reducing buoyancy. They also claimed that the delayed raindrop production in
the continental cloud increases the duration of the diffusion droplet growth stage,
which in turn, increases the latent heat release by condensation.
Seigel (2014) showed an increase in $w$ with increasing aerosol loading in the cloud
core in numerical simulations of a warm convective cloud field. He also showed a



decrease in cloud size under polluted conditions due to increased mixing between the
clouds and their dry environment.
It has been recently shown (Dagan et al., 2015a;Dagan et al., 2015b;Dagan et al.,
2017) that under given environmental conditions, warm convective clouds have an
optimal aerosol concentration ($N_{\_op}$) with respect to their macrophysical properties
(such as total mass and cloud top height) and total rain yield. For concentrations
smaller than $N_{\_op}$, the cloud can be considered as aerosol-limited (Koren et al.,
2014;Reutter et al., 2009), and a positive correlation between the aerosol
concentration and cloud properties can be expected. Suppressive processes such as
enhanced entrainment and water loading take over when the concentrations are higher
than $N_{\_op}$ and reverse the trend. It has also been shown that the value of $N_{\_op}$ depends
heavily on the environmental conditions (thermodynamic conditions that support
deeper clouds would have a larger $N_{\_op}$).
In this work, a bin-microphysics cloud model and large eddy simulation (LES) of a
cloud field were used to explore how changes in aerosol concentration affect $w$ and $\eta$,
the interplay between them and, as a result, the height of the COG in warm convective
clouds (Koren et al., 2009;Grabowski et al., 2006).

**2. Methodology**
**2.1 Single-cloud model**
The Tel Aviv University axisymmetric nonhydrostatic cloud model (TAU-CM) with
detailed treatment of cloud microphysics (Reisin et al., 1996;Tzivion et al., 1994) was
used. The included warm microphysical processes were nucleation of droplets,
condensation and evaporation, collision–coalescence, breakup, and sedimentation.
The microphysical processes were formulated and solved using a two-moment bin
method (Tzivion et al., 1987).
The background aerosol size distribution used here represents a clean maritime
environment (Jaenicke, 1988). The aerosols are assumed to be composed of NaCl.
The different aerosol concentrations (25, 500 and 10,000 cm$^{-3}$, denoted hereafter as
25CCN, 500CCN and 10000CCN, respectively) and size distributions are identical to



124 those used in Dagan et al. (2015a). To study the involved processes, we used a wide

125 range of aerosol loading conditions, from extremely pristine to extremely polluted. To

126 avoid giant CCN effects, the aerosol size distribution was cut at 1 µm (Feingold et al.,

127 1999;Yin et al., 2000;Dagan et al., 2015b).

128 The model resolution was set to 50 m, in both the vertical and horizontal directions,

129 and the time step to 1 s. The initial conditions were based on theoretical atmospheric

130 profiles that describe a tropical environment (Malkus, 1958) (see profile T1RH2 in

131 Fig. 1 in Dagan et al., 2015a). They consisted of a well-mixed sub-cloud layer

132 between 0 and 1000 m, a conditionally unstable cloudy layer (6.5°C/km) between

133 1000 and 4000 m, and an overlying inversion layer (temperature gradient of 2°C over

134 50 m). The relative humidity (RH) in the cloudy (inversion) layer was 90% (30%).

135 The results presented here were examined for a few different sets of initial conditions

136 (different inversion-base heights and RH in the cloudy layer—analysis not shown).

137 The general conclusions were found to be insensitive to the initial conditions.

138

139 To examine the effect of aerosols on the entire cloud, the properties presented in this

140 work are cloud mean values weighted by the liquid water mass in each grid cell.

141 Cloudy grid cells were defined as cells with liquid water content larger than 0.01 g/kg.

142 The cloud's COG (Koren et al., 2009;Grabowski et al., 2006) was calculated as:

$$COG = \frac{\Sigma m_i z_i}{\Sigma m_i}$$

143  (1)

144 where $m_i$ and $z_i$ are the mass [kg] and height [m] of voxel $i$, respectively.

145 The $\eta$ (effective terminal velocity) was calculated according to Koren et al. (2015):

$$\eta = \frac{\Sigma Vt_j m_j n_j}{\Sigma m_j n_j}$$

146  (2)

147 where $Vt_j$, $m_j$ and $n_j$ are the terminal velocity [m/s], mass [kg] and concentration [cm$^{-3}$]

148 of droplets in bin $j$, respectively. This was calculated for all cloudy grid cells.







**2.2 LES**
We used the System for Atmospheric Modeling (SAM) LES model (Khairoutdinov
and Randall, 2003) with a bin-microphysics scheme (Khain and Pokrovsky, 2004) to
simulate the BOMEX (Barbados Oceanographic and Meteorological EXperiment)
warm cumulus case study (Holland and Rasmusson, 1973;Siebesma et al., 2003). The
horizontal resolution was set to 100 m, the vertical resolution to 40 m. The domain
size was 12.8 x 12.8 x 4.0 km$^3$ and the time step was 1 s. We ran the model for 16 h,
but the statistical analysis included only the last 14 h of the simulation. We used 8
different aerosol concentrations: 5, 25, 50, 100, 250, 500, 2000 and 5000 cm$^{-3}$. Again,
we used a marine background aerosol size distribution (Jaenicke, 1988). Further
details about the simulations can be found in Dagan et al. (2017).

**3. Results and discussion**
**3.1 Single cloud: vertical velocity and effective terminal velocity**
Starting from the single-cloud scale, we first followed mean $w$, mean $\eta$, mean $V_{COG}$,
and COG height as a function of time for the three different levels of aerosol loading
(25, 500, and 10,000 cm$^{-3}$). From an early stage of the cloud's evolution, the cleanest
cloud (25CCN) had the lowest COG. This was a result of the lower $w$ (Fig. 1a) and
larger absolute value of the negative $\eta$ (caused by the initially larger droplets – Fig.
1b), which together cause a lower $V_{COG}$ (Fig. 1c). At the early stages of the polluted
clouds, the 500CCN and 10000CCN COG moved upward at the same rate. After
about 60 min of simulation, the 500CCN's COG started to decrease while the
10000CCN's COG remained relatively high. This trend could not be explained by the
cloud's mean $w$ (Fig. 1a). The 500CCN's $w$ was higher than that of the 10000CCN
during the period between 50 and 63 min of simulation. Without considering the
effect of $\eta$ on the COG, one would expect that the 500CCN's COG would be higher
than that of the 10000CCN. The 500CCN had lower (more negative) values of $\eta$ than
the 10000CCN, which decreased the height of its COG compared to the 10000CCN.
These larger negative values of $\eta$ in the 500CCN were due to the rain that developed
from this cloud (the rain from the 10000CCN is negligible), which led to lower
mobility (lower ability to move with the ambient air (Koren et al., 2015)).





Figure 1 demonstrates the importance of the aerosol effect on both $w$ and $\eta$ in
determining the COG height. Figure 2 presents the evolution of the clouds on the
phase space span by $w$ vs. $V_{COG}$. All clouds began their evolution on the 1:1 line. This
means that at the early stages of the cloud's evolution, $\eta \sim 0$ and hence $V_{COG} \sim w$.
After about 40 min of simulation, the cleanest cloud's (25CCN) trajectory began to
deviate from the 1:1 line to the left, demonstrating an increase in $|\eta|$ and hence lower
droplet mobility. The deviation from the 1:1 line occurred later (at about t = 55 min of
simulation) in the more polluted simulation (500CCN), whereas for the most polluted
clouds (10000CCN), the lack of significant collision–coalescence and rain production
resulted in evolution on the 1:1 line throughout the cloud's lifetime. This delay in the
deviation from the 1:1 line (increasing the time for which $\eta \sim 0$) demonstrates the
increase in droplet mobility with aerosol loading. The longer period for which $\eta \sim 0$ in
the polluted cases enables the water mass to be pushed higher into the atmosphere and
hence (together with the increase in the air vertical velocity – Fig. 1a) to produce
cloud invigoration by the aerosol (Koren et al., 2015).

**3.2 LES results: aerosol effect on the vertical velocity and effective terminal**
**velocity in cloud fields**
Shifting our view from the single-cloud scale to the cloud-field scale adds another
layer of complexity as clouds affect the way in which the whole field's
thermodynamics evolve with time. Aerosol concentration has recently been shown to
determine the trend of this evolution (Dagan et al., 2016;Dagan et al., 2017). Clean
precipitating clouds act to consume the initial instability that created them by
warming the cloudy layer (in which there is net condensation) and cooling the sub-
cloud layer (by rain evaporation). On the other hand, polluted non-precipitating clouds
act to increase the field's instability by cooling and moistening the upper cloudy and
inversion layers.
Figure 3 presents the domain's mean $w$ (in both space and time, weighted by the liquid
water mass to be consistent with the COG view) vs. the domain mean $\eta$. The color-
coding in Fig. 3 denotes the different aerosol concentrations. In agreement with
previous studies (Saleeby et al., 2015;Seigel, 2014), an increase in aerosol loading
yielded an increase in $w$. This increase is driven by larger latent heat contribution to




the cloud's buoyancy due to the increased condensation efficiency (Dagan et al.,
2015a;Dagan et al., 2017;Koren et al., 2014;Pinsky et al., 2013;Seiki and Nakajima,
2014) and thermodynamic instability (Dagan et al., 2016;Dagan et al., 2017). In
parallel, aerosol shifts to smaller droplets (Squires, 1958) and reduces the magnitude
of $\eta$, indicating better mobility of the smaller droplets (Koren et al., 2015). The
outcome of these two effects (that work together to push the water mass higher in the
atmosphere) is an increase in COG height with aerosol loading (Heiblum et al.,
2016b;Dagan et al., 2017).
In the single-cloud-scale analysis (section 3.1), we showed how the timing of the
evolution of the two velocities dictates the aerosol effect. Here, having many clouds in
the field in different stages of their lifetimes, we first analyzed the bulk properties of
the two velocities. With the intention of quantifying the relative contribution of the
aerosol effect on the mean COG height by modulating $w$ and $\eta$, we plotted them one
against the other for all of the simulations that differed in aerosol loading and for all
clouds in the domain (Fig. 3a). For the entire simulation period, the $\eta$ vs. $w$ scatter
plot resulted in an almost a straight line ($R^2 = 0.96$) which was sorted by aerosol
concentration with a slope of 0.69. This means that an increase in aerosol
concentration that will result in a 1 m/s increase in mean $w$ will drive a decrease in the
magnitude of $|\eta|$ by 0.69 m/s. In other words, the relative contribution to the changes
in the mean COG height in the domain caused by the increase in aerosol loading
(Heiblum et al., 2016b;Dagan et al., 2017) during the entire simulation is ~60% due to
changes in $w$ and ~40% due to changes in $\eta$.
To include the aerosol effect on the cloud-field-scale thermodynamic properties, we
divided the simulation periods into three equal thirds (excluding the first 2 h, each
third of a period covered 4 h and 40 min). The x and * markers in Fig. 3a represent the
first third (2 h to 6 h 40 min into the simulation) and last third (11 h 20 min to 16 h
into the simulation), respectively. During the first third, the slope of $w$ vs. $\eta$ was
steeper than the mean over the entire simulation (slope of 0.92 with $R^2 = 0.96$); during
the last third, it was more gradual (slope of 0.47 with $R^2 = 0.87$). The almost 1:1
relation between $w$ and $\eta$ in the first third of the simulation period suggests a
comparable contribution in determining the aerosol effect on mean COG height.
However, the relative contribution of $\eta$ decreases as the simulation progresses, to
about 1/3 during the last third of the simulation period (compared with 2/3 of $w$).





The decrease in the $w$ vs. $\eta$ slopes toward the end of the simulations is driven by the
changes in the thermodynamic instability. The increase in instability under polluted
conditions produces an increase in mean $w$ (Dagan et al., 2016). Nevertheless,
increased instability and deepening of the cloud layer are not sufficient to produce a
significant amount of rain under the most polluted simulations and hence, there is no
increase in the magnitude of $\eta$. An increase in $w$ with no change in $\eta$ is manifested as
a horizontal shift to the right on the $\eta$ vs. $w$ phase space (red arrow in Fig. 3a). On the
other hand, the decreased instability under clean conditions produces a decrease in
both mean $w$ and the rain amount (Dagan et al., 2017), and therefore in $|\eta|$ (blue arrow
in Fig. 3a). The end result of the different changes in $w$ and $\eta$ under clean and polluted
conditions is a decrease in the slope of $\eta$ vs. $w$ and therefore, a decrease in the relative
contribution of $\eta$ to the aerosol effect on the mean COG.

In Fig. 3a, the presented quantities are domain and time averages. Figure 1 showed
that the relative contribution of $w$ and $\eta$ to the aerosol effect on COG height strongly
depends on the stage of the cloud's evolution. The averaging in Fig. 3a mixes many
clouds at different stages in their evolution and represents the effect on the mean COG
in the domain. To further explore the relative contribution of the aerosol effect on $w$
and $\eta$ as a function of cloud-evolution stage, we used a cloud-tracking algorithm
(Heiblum et al., 2016a). We identified the growing stage of the clouds as the stage for
which the cloud top ascends. Figure 3b presents the $\eta$ vs. $w$ phase space only for
clouds in their growing stage. Table 1 presents the slopes of the linear regression lines
for the entire simulation time and for the different thirds of the simulation period. The
decrease with time in the relative contribution of $\eta$ compared to $w$ to the aerosol effect
on COG height was also seen for the growing clouds (see the decrease in the slope
with time). This, again, was due to the changes in thermodynamic conditions.
As shown for the cloud scale, one of the most notable aerosol effects can be viewed as
delaying the onset of significant collection processes in the polluted clouds (Koren et
al., 2015), and therefore delaying the increase in $\eta$ values early in the cloud's lifetime.
Therefore, during the growing stage, the relative contribution of $\eta$ was higher (Fig.
3b) as compared to "all clouds" (Fig. 3a). This was demonstrated by the increasing
slope of the $\eta$ vs. $w$ phase space during the growing stage (Table 1).





To quantify the evolution of the thermodynamic instability with time as a function of
aerosol loading, we looked at the time trends in the $\eta$ vs. $w$ phase space. We defined
the angle '$A$' as the angle between the time trend points on the $\eta$ vs. $w$ phase space per
given aerosol loading (the line that connects the first and last thirds of the simulation
and the x-axis on the $\eta$ vs. $w$—see schematic definition of $A$ in Fig. 3b). We note that
$A$ rotates counter-clockwise with increasing aerosol loading (Fig. 3a). It starts as
~100º for the cleanest simulation and monotonically increases with aerosol loading to
~360º for the most polluted simulations (Fig. 4b). $A$ between 90º and 180º (as shown
for clean cases—Fig. 4b) represents a decrease in both $w$ and $|\eta|$ and hence a decrease
in the thermodynamic instability with time. $A$ between 270º and 360º, on the other
hand (as shown for the most polluted cases—Fig. 4b), represents an increase in both $w$
and $|\eta|$ and hence an increase in the thermodynamic instability with time.
The sign of $V_{COG}$ has been shown to predict the evolution of thermodynamic
instability (Dagan et al., 2016). Thus, correlations between $A$ and $V_{COG}$ are expected.
Figure 4 presents $V_{COG}$ (Fig. 4a) and $A$ (Fig. 4b) as a function of the aerosol loading,
and $V_{COG}$ vs. $A$ (Fig. 4c). Figure 4a and b demonstrates that both the $V_{COG}$ and $A$
increase monotonically with aerosol loading following a similar trend. $V_{COG}$ and $A$
cross the 0 and 180º lines, respectively, at similar aerosol concentrations, representing
the transition between consumption and production of the thermodynamic instability
(Dagan et al., 2016). Figure 4c further demonstrates an almost perfect linear
correlation ($R^2 = 0.99$) between $V_{COG}$ and $A$ sorted by aerosol concentration.

**3.3 Summary**
Clouds form a complex system in which microphysical and dynamical processes are
tightly linked and modulated by the thermodynamic properties of the environment. In
turn, on the cloud-field scale, clouds affect the field's thermodynamic conditions. The
aerosol effect on droplet size distribution therefore affects all of the above.
Analyzing the two characteristic velocities on the cloud scale allows separation, as a
first approximation, between the aerosol effects on condensation/evaporation
efficiencies (reflected by the magnitude of $w$) and those on droplet mobility (reflected





by the inverse magnitude of $\eta$). The magnitudes of $w$ and $\eta$ act in opposite ways, i.e.,
stronger $w$ and smaller $|\eta|$ imply more efficient transport of liquid water to the upper
atmosphere. We use their sum, defined as $V_{COG}$, to estimate the overall effect on the
COG's vertical movement. Single-cloud analysis showed the timing of this interplay
and how each velocity affects the COG elevation. It showed that the invigorating
aerosol effect can be viewed mostly at the early stages of cloud development, when an
increase in aerosol loading enhances the condensation efficiency (reflected as higher
$w$ levels) and delays the onset of significant collection processes (reflected as a delay
in the sharp increase in $\eta$). Both act to transfer liquid water higher into the atmosphere
(Koren et al., 2015). Later, as the cloud dissipates, the "payment" is viewed as
enhanced evaporation, and if the cloud manages to reach the significant collection-
process stage, then the surface rain is stronger (expressed as a sharp increase in $|\eta|$).
Similar to the single-cloud case, the LES results demonstrated an increase in $w$ and
decrease in the magnitude of $\eta$ (less negative $\eta$) with aerosol loading, both yielding a
higher COG. We analyzed the bulk properties of the two velocities for the entire
simulation time (14 h) and for all clouds in the domain and showed that the relative
contribution of the aerosol effect on $w$ and $\eta$ in determining COG evolution is
comparable (60% and 40%, respectively). However, at the beginning of the
simulation, this ratio was almost 1:1, and the relative contribution of $\eta$ decreased with
time. Such temporal changes in the $w$ vs. $\eta$ slope indicate changes in the
thermodynamic properties of the field (Dagan et al., 2016). Increasing thermodynamic
instability under polluted conditions results in an increase in mean $w$, while the
decreasing instability under clean condition results in a decrease in rain amount and
hence, in $\eta$. Both trends act to reduce the slope.
Using a cloud-tracking algorithm, we identified the growing stage of the clouds and
examined the relative contribution of the aerosol effect on COG height by modulating
$w$ and $\eta$ during this stage. We showed that the relative contribution of the aerosol
effect on $\eta$ is larger during the growing stage (for which aerosol loading acts to
maintain lower $|\eta|$ for a longer time) compared to the mature and dissipating stages,
thereby strengthening the argument that most of the aerosol invigoration effect occurs
early in the cloud's evolution (Koren et al., 2015).




*Data availability.* Information about the model and initialization files are available upon request to the contact author.

*Competing interests.* The authors declare that they have no conflict of interest.

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



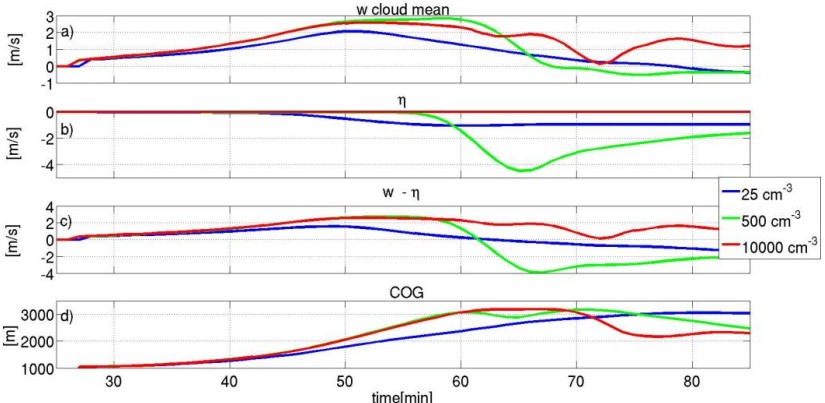


**Figure 1. (a) Mean vertical velocity ($w$) , (b) mean effective terminal velocity ($\eta$),**

**(c) mean vertical velocity plus effective terminal velocity, and (d) cloud center of**

**gravity (COG) as a function of time for three different aerosol concentrations.**


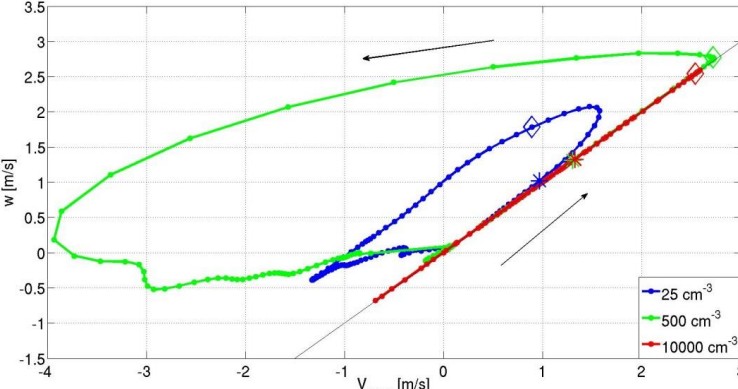


**Figure 2. Cloud evolution on the phase space span by $w$ vs. $V_{COG}$. The arrows**

**mark the direction of the trajectories and the thin black line is the 1:1 line. Stars**

**and diamonds denote t = 40 min and 55 min of the simulation, respectively.**




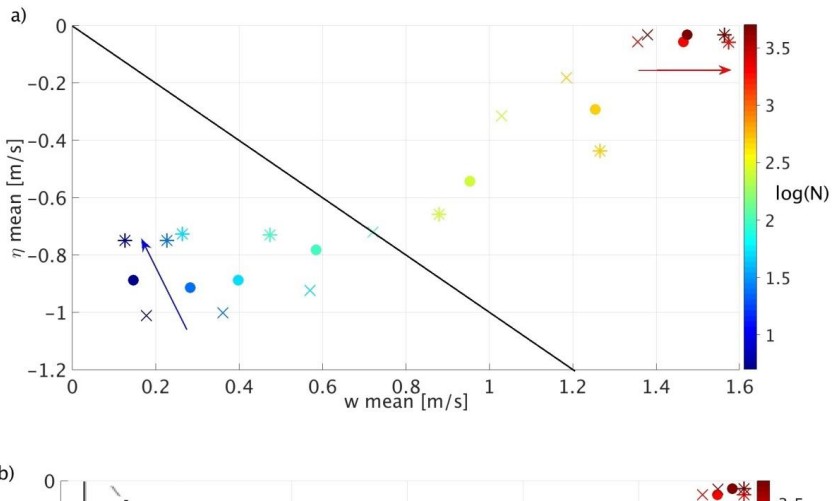

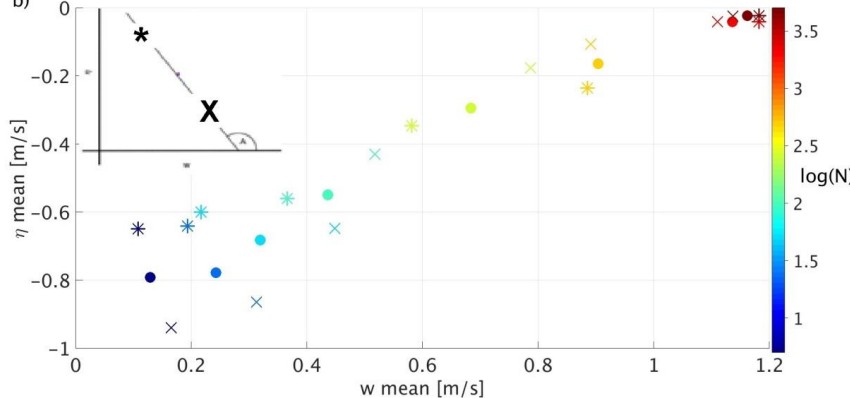


**Figure 3. Temporal and spatial averages of the ambient air vertical velocity ($w$)**
**vs. effective terminal velocity ($\eta$). Color-coding denotes the different aerosol**
**concentrations. Dots represent averages of the entire simulation data (excluding**
**the first 2 h spin-up time). The x and * markers represent the first third (2 h to 6**
**h 40 min) and last third (11 h 20 min to 16 h) of the simulation period,**
**respectively. (a) All clouds in the domain. (b) Only clouds in the growing stage.**
**The black line in (a) is the zero-sum line for which $V_{COG} = 0$ (below the line $V_{COG}$**
**$< 0$ and above it $V_{COG} > 0$). The angle A that measures the $\eta$ vs. $w$ time trend per**
**aerosol level is illustrated in the inset in panel b.**

492




**Table 1. Linear regression slope on the $\eta$ vs. $w$ phase space for the different periods of the simulations for all clouds and growing-stage clouds in the domain. $R^2$ of the regression lines is presented in parentheses.**

|  | All clouds | Growing clouds |
|---|---|---|
| **Total simulation period (2–14 h)** | 0.69 (0.96) | 0.79 (0.98) |
| **First period of simulation (2–6:40 h)** | 0.92 (0.96) | 0.99 (0.93) |
| **Last period of simulation (11:20–16 h)** | 0.47 (0.87) | 059 (0.98) |

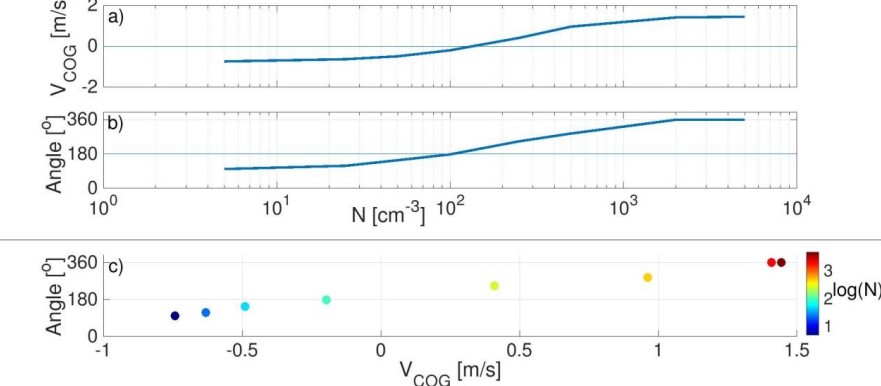

**Figure 4. (a) The cloud field's mean value of $V_{COG}$ and (b) the angle $A$ between the line that connect the first and last thirds of the simulation period and the x-axis on the $\eta$ vs. $w$ phase space for all clouds in the domain (Fig. 3a) as a function of aerosol loading. (c) $V_{COG}$ vs. $A$. Color-coding denotes the different aerosol concentrations.**