# Peer review of "Quantifying the effect of aerosol on vertical velocity and effective terminal velocity in warm convective clouds Guy Dagan, Ilan Koren\*, and Orit Altaratz"

_Atmospheric Chemistry and Physics, 2018_

## Referee Comment (RC1) · Anonymous Referee #1 · 25 Mar 2018

Overview

This study investigated the aerosol effects on warm convective clouds via the interplay of updraft (w) and effective terminal velocity (eta) using both single cloud model and large eddy simulation (LES). The topic is very interesting and the authors provided detailed analysis on how aerosols affect the evolution of thermodynamic instability of cloud field using both characteristic vertical velocities.

General comments

1. The simulated clouds from both cloud model and LES were growing at very humid environment, e.g., "the relative humidity in the cloudy layer was 90%" (L134). The au-

thors found monotonically increased vertical velocity with enhanced aerosol loading, i.e., aerosol invigoration effect on warm convective clouds. As we know from previous studies, however, the aerosol indirect effect varied with different atmospheric conditions. For instance, if the atmosphere is not so humid or the relative humidity is lower than the current study, will the aerosol suppress effect be expected? If so, how will aerosol effects on both characteristic vertical velocities as well as thermodynamic properties change with varied relative humidity?

Specific comments:

1. Fig. 4 show nicely the similar trend of COG velocity (VCOG) and the "authors-defined-Angle" (A). The authors argued that the VCOG can be an indicator of thermodynamic instability and also of the aerosol effects on warm clouds. But for A, what can we learn from the evolution of it? Or does it have different function with VCOG? The amplitude of A seems depend on the relative amplitude of both velocities between the first and the last third part of simulation period, in other words, the order of peak time of both characteristic velocities for different aerosol scenarios. If w and (eta) peak earlier (at the first third stage in a clean run, for example) during the cloud lifetime, A might be around ∼100 degree; whereas if they peak later (say at the last third part and may appear in polluted run), A might be around ∼360 degree. I hope the authors give more information about how we can use the parameter A in the study of aerosol-cloud interactions.

2. The authors split the simulation period into 3 parts for the analysis of LES runs in Fig. 3a. Does the three parts stand for different stages of cloud evolution?

And in Fig. 3b, "only clouds in the growing stage" were considered for the first and the last third part of simulation period. I am confused here. What should we expect for growing clouds at different part of simulation period?

3. Fig. 1c and 1d show that VCOG became negative after 65 min for clean run (blue curve), but COG height continues to increase until the end of the simulation. What is

Interactive
comment

the reason for that, since negative VCOG seems denote the downward movement of center of gravity of cloud?

4.L275: "and therefore delaying the increase in (eta) values early in the cloud's lifetime", I think (eta) should be |eta|. The value of (eta) is originally negative. I suggest the authors check the descriptions of |eta| and (eta) in the manuscript carefully.

---

## Referee Comment (RC2) · Anonymous Referee #2 · 13 Apr 2018

Review of the study "Quantifying the effect of aerosol on vertical velocity and effective terminal velocity in warm convective clouds", authored by Guy Dagan, Ilan Koren, and Orit Altaratz

The study is dedicated to analysis of general properties of a warm cumulus clouds on aerosol loading. The analysis is performed in the terms of Center of Gravity (COG) behavior. It is shown that aerosols induce cloud invigoration, and that the effects are maximum at the developing cloud stage.

The paper is of interest. The approach to the analysis is original. The study is recommended to publication with minor revisions.

[Figure]

The comments and remarks are listed below.

Line 13. Which processes do you mean? Trends of which? Be more specific.

line 75. Clarify, what is "clean precipitation conditions".

line 94. Why does mixing increase? Seigel (2014) explains this effect by stronger evaporation of smaller droplets in polluted cases.

line 99. Do you mean accumulated rain at the surface?

line 102. The sentence is not clear. What does a "correlation between aerosols and cloud properties" mean? Be more specific.

Line 110. What do you mean presenting the references? How is the study by Grabowski et al., 2006 related to the COG behavior?

line 115. Can you comment the choice of the model? The axisymmetric model may have problem because rain mass (mass loading) is located in the cloud center substantially decreasing updrafts. Such configuration may decrease the generality of the conclusions, because even small wind shear may form a moment between updrafts caused by buoyancy and downdrafts caused by mass loading and other factors.

line 119. Tzivion et al. refer the method they developed to as microphysical method of moments (MMM). The name "two-moment bin method" is somehow confusing, because alternative SBM method used in SAM calculates size distribution functions and the values of any moment of DSD in each bin.

line 135. The sentence is not clear. Results certainly should depend on inversion-base heights, and RH in cloud layer.

line 151. The title is not suitable, in my opinion. The simulations with a single cloud model discussed in section 2.1 is also LES. The authors, supposedly, want to stress that in SAM they simulate cloud

line 165. How were the mean values calculated? Over entire cloud?

line 196. Can you comment the potential effect of the fact that the model used is axisymmetric? The effect of wind shear should be very significant.

line 209. Please clarify what is "weighted by the liquid water mass". Please present expression used for the calculation.

lines 213-214. Saleeby et al. and Seigel use RAMS bulk parameterization scheme. Do you suppose that this scheme describes increase in the latent heat release by the increase in aerosol loading?

Conclusion section. It would be important to add a discussion about the applicability of the results to evaluation of aerosol effects on radiative cloud properties, on precipitation amounts, etc.

---

## Author Comment (AC1) · 26 Apr 2018

Reply to reviewers' comments:

**Quantifying the effect of aerosol on vertical velocity and effective terminal velocity in warm convective clouds**

We would like to thank the reviewers for their insightful and helpful comments that helped us clarify and improve this work. Please find below a point-by-point reply to all of the reviewers' comments (in blue)

**Reviewer 1:**

This study investigated the aerosol effects on warm convective clouds via the interplay of updraft (w) and effective terminal velocity (eta) using both single cloud model and large eddy simulation (LES). The topic is very interesting and the authors provided detailed analysis on how aerosols affect the evolution of thermodynamic instability of cloud field using both characteristic vertical velocities.

**Answer:** We thank the reviewer for the important comments and we are happy that the reviewer found this work interesting.

**General comments:**

1. The simulated clouds from both cloud model and LES were growing at very humid environment, e.g., "the relative humidity in the cloudy layer was 90%" (L134). The authors found monotonically increased vertical velocity with enhanced aerosol loading, i.e., aerosol invigoration effect on warm convective clouds. As we know from previous studies, however, the aerosol indirect effect varied with different atmospheric conditions. For instance, if the atmosphere is not so humid or the relative humidity is lower than the current study, will the aerosol suppress effect be expected? If so, how will aerosol effects on both characteristic vertical velocities as well as thermodynamic properties change with varied relative humidity?

Answer: Thank you for this important comment. This question was in the focus of several of our recent studies. Indeed aerosol effect on clouds depends on the environmental conditions (like RH and instability) as was shown in previous studies

((Fan et al., 2009;Seifert and Beheng, 2006;Khain et al., 2008;Lee et al., 2008) and in additional studies that focused on warm convective clouds- (Dagan et al., 2015a;Dagan et al., 2015b;Sato et al., 2018)). This dependence on the thermodynamic conditions can be explained by a competition between core and margin dominated processes, in which core processes have an adiabatic nature whereas margin processes are controlled by mixing and entrainment. In warm convective clouds the fuel for invigoration relies on core-oriented processes, i.e. enhanced condensation in elevated aerosol concentration conditions that increase the updraft and transport the smaller droplets (that have smaller effective terminal velocity) higher in the atmosphere. While clouds suppression processes are mostly periphery processes, i.e. enhanced evaporation that fuels stronger downdrafts and mixing (in addition to the suppressive effect of the water loading). Core processes take place within the core volume while margin processes occur on the interface between cloud and the ambient air. Therefore, in relatively small clouds for which the surface area to volume ratio is relatively large, entrainment will be a dominant process. On a similar manner, since entrainment is controlled by RH gradients, in drier conditions the entrainment is expected to be stronger. Therefore, in lower RH conditions or under initial conditions that support shallower clouds (for example under lower inversion base height) margin processes are enhanced as the aerosol concentration increases and therefore cloud suppression is expected to take place above a turning point that corresponds to lower aerosol concentration (lower optimal aerosol concentration) (Dagan et al., 2015a;Dagan et al., 2015b).

The important point for this comment is that the turning point between invigoration and suppression (the optimal aerosol concentration), and the slopes of those opposite trends are different when examining different clouds' properties. Cloud's properties that are more "core-oriented" (see examples below) will be more positively affected by aerosols. While cloud's properties that are more "periphery-oriented" will be more negatively affected by aerosols (Dagan et al., 2015a;Dagan et al., 2017).

The mass weighted mean air vertical velocity (w) is a "core-oriented" property because of two reasons: 1) the strongest air vertical velocities are found in the cloud core, and 2) the averaging by mass gives more weight to the more massive core than to the more diluted periphery (which contain, on average, lower liquid water content). Hence, even under drier or more stable initial conditions, *w* analysis will reflect more the core invigoration effect and will show a turning point at high aerosol concentrations. This can be seen in Fig. 7 taken from Dagan et al. (2015a) (shown below) which presents the maximum over time of w as a function of aerosol concentration for nine different sets of initial conditions spreading a wide range of environmental conditions as expressed by various relative humidity values and cloudy layer depths. It demonstrates that the invigoration part of the trend is much more significant than the suppression part for w. This is true compared with the other clouds' properties such as the total surface rain yield which is affected by the entire cloud and hence by both the cloud's core and periphery. Please note that the results presented in the manuscript demonstrate this behavior: the turning point corresponds to high aerosol loading with less significant suppression trend compared to the invigoration part of the trend (Fig. 1 in the manuscript shows that the 10000CCN simulation has a bit lower w than the 500CCN but it is higher than in the 25CCN simulation).

Figure 7. The cloud's maximum top height (top panels), the maximum over time of the mean vertical velocity weighted by the mass in each grid point (middle panels) and the total surface rain yield (bottom panels) as a function of the aerosol loading, for each simulated cloud as a function of the aerosol concentration. Each curve represents 10 simulations performed for an initialization profile (a total of 9 profiles). T1 represents a profile with an inversion layer located at 4 km, T2 at 3 km, and T3 at 2 km. RH1 represents a profile with 95% RH in the cloudy layer, RH2-90%, and RH3-80%. Taken from Dagan et al. (2015a).

To demonstrate that our simulations results are general we have conducted an additional set of simulations with drier initial conditions (RH=80% instead of 90%, as presented in the paper) and shallower cloudy layer (1 km instead of 3 km). This initial atmospheric profile is denoted as T3RH3 in Dagan et al. (2015a) and the results are presented in Fig. R1, below. In this shallower and drier cloudy layer the turning point between invigoration to suppression trend corresponds to lower aerosol concentration (Dagan et al., 2015a). However, even in this case (as in the case presented in the manuscript), the *w* of the two polluted clouds (500 cm-3 and 10000 cm-3) initially increase at the same rate while the clean cloud has lower *w*. The maximum *w* over time is higher for the most polluted cloud (10000 cm-3 - 2.16 m/s) compared to the cleanest cloud (25 cm-3 - 1.98 m/s) representing less significant role of the suppression part in the trend as compared to the invigoration part for *w*.

In this shallower and drier cloudy layer the aerosol concentration above which there is total rain suppression is lower than 500 cm-3 (see Fig. 7 above and Dagan et al. (2015b)). Thus, there is a sharp increase in  $/\eta/$  (representing a significant amount of precipitation) only in the cleanest cloud (25 cm-3) for this initialization profile. However, during the growing stage of the clouds,  $/\eta/$  demonstrates an increase in the droplet mobility also in this case. For example, after 40 min of simulation  $\eta = -7.7 \cdot 10^{-4}$ ,  $-8.2 \cdot 10^{-3}$  and  $-5.3 \cdot 10^{-2}$  m/s for the simulations with the aerosol loading of 10000, 500 and 25, respectively.